# Requirements of an Application to Monitor Diet, Physical Activity and Glucose Values in Patients with Type 2 Diabetes: The Diameter

**DOI:** 10.3390/nu11020409

**Published:** 2019-02-15

**Authors:** Niala den Braber, Miriam M. R. Vollenbroek-Hutten, Milou M. Oosterwijk, Christina M. Gant, Ilse J. M. Hagedoorn, Bert-Jan F. van Beijnum, Hermie J. Hermens, Gozewijn D. Laverman

**Affiliations:** 1Department of Internal Medicine/Nephrology, Ziekenhuisgroep Twente (ZGT), 7609 PP Almelo, The Netherlands; m.m.r.hutten@utwente.nl (M.M.R.V.-H.); mi.oosterwijk@zgt.nl (M.M.O.); cm.gant@meandermc.nl (C.M.G.); i.hagedoorn@zgt.nl (I.J.M.H.); g.laverman@zgt.nl (G.D.L.); 2Biomedical Signals and Systems (BSS), University of Twente, 7522 NB Enschede, The Netherlands; b.j.f.vanbeijnum@utwente.nl (B.-J.F.v.B.); h.j.hermens@utwente.nl (H.J.H.); 3Roessingh Research and Development (RRD), 7522 AH Enschede, The Netherlands

**Keywords:** Type 2 diabetes mellitus, diabetes management, dietary application, dietary assessment, nutrition, physical activity, blood glucose, mHealth

## Abstract

Adherence to a healthy diet and regular physical activity are two important factors in sufficient type 2 diabetes mellitus management. It is recognized that the traditional treatment of outpatients does not meet the requirements for sufficient lifestyle management. It is hypothesised that a personalized diabetes management mHealth application can help. Such an application ideally measures food intake, physical activity, glucose values, and medication use, and then integrates this to provide patients and healthcare professionals insight in these factors, as well as the effect of lifestyle on glucose values in daily life. The lifestyle data can be used to give tailored coaching to improve adherence to lifestyle recommendations and medication use. This study describes the requirements for such an application: the *Diameter*. An iterative mixed method design approach is used that consists of a cohort study, pilot studies, literature search, and expert meetings. The requirements are defined according to the Function and events, Interactions and usability, Content and structure and Style and aesthetics (FICS) framework. This resulted in 81 requirements for the dietary (*n* = 37), activity and sedentary (*n* = 15), glycaemic (*n* = 12), and general (*n* = 17) parts. Although many applications are currently available, many of these requirements are not implemented. This stresses the need for the *Diameter* as a new personalized diabetes application.

## 1. Introduction

Type 2 Diabetes Mellitus (T2DM) is one of the most common chronic diseases, which was recorded to affect 415 million people worldwide in 2015. Its prevalence is still increasing due to the rise of obesity and unhealthy lifestyle with an expected prevalence of 642 million in 2040 [1,2].

The two main components of a sufficient diabetes management are adherence to a healthy diet and regular physical activity. These lifestyle components are important for both glycaemic control, i.e., keeping blood glucose levels within the target range, and in maintaining long-term health, which includes the prevention of micro- and macrovascular complications [3]. The risk of developing complications increases with a poor diet quality and insufficient physical activity [4,5,6].

For many T2DM patients, the everyday challenge includes the restriction of carbohydrate intake and the performance of sufficient physical activity while reducing sedentary behaviour. Another challenge is to adhere to, often a plethora of, pharmacological agents. In patients that are treated with insulin, there is the additional burden of insulin injections and the need to monitor blood glucose. Multiple finger-pricks per day, in combination with an estimation of the carbohydrate content of the meal, are usually necessary to determine the appropriate pre-meal insulin dose [7,8].

Diabetes healthcare professionals (HCPs) support patients, whereby they provide education and coaching on lifestyle and pharmacological therapy [9]. However, in the current situation, regular contacts between patients and professionals are particularly suited to monitor the pharmacological management of glycaemic control, blood pressure, and dyslipidaemia. It is increasingly recognized that the traditional set-up of outpatients does not meet the requirements for sufficient lifestyle management, and therefore efforts for the improvement of lifestyle and self‒management do not reach full potential [10,11]. To illustrate this, we found in the vast majority of T2DM patients that were treated in our hospital, “ZiekenhuisGroep Twente” (ZGT), where adherence to the guidelines of physical activity and healthy diet was not met. This was reflected by an average body mass index (BMI) of 33 kg/m^2^ and a sufficient vegetable intake in only 7% of the patients [12,13]. Of note, lifestyle behaviour is not measured routinely and objectively in clinical practice. However, objective measurements are of great importance in adequately mapping lifestyle, because patients grossly overestimate their healthy lifestyle behaviours [14].

Advances in technology have made it easier to monitor lifestyle, e.g., through smartphone applications and wearable technology. Worldwide, several initiatives have arisen to transform healthcare and health support [15,16,17]. The use of technology can help to incorporate effective lifestyle management in routine clinical care. Mobile health (mHealth) technology allows for lifestyle parameters to be monitored objectively and continuously [15]. Objective data provides insight in actual lifestyle habits, both to the patient and the HCP, while also increasing awareness [18,19]. Additionally, data regarding carbohydrate- and fat intake, physical activity, medication use, and glucose values can be combined, which is of importance because these factors influence each other [19,20]. When compared to traditional methods of obtaining insight in patients’ lifestyle, technology-based methods are ultimately less expensive, can be used when it suits the patient, and provide objective data. Moreover, technology can be used for (digital) tailored coaching [21]. Technologies that measure nutrition, physical activity, or provide glucose values are already available, but there are major limitations [18]. The existing applications are usually developed for personal use rather than for clinical use; no application exists that can measure and integrate all of the aspects that are considered to be necessary for an optimal diabetes management, and coaching functionalities remain only rudimentary [22].

Therefore, our research group aims to develop a digital tool that incorporates lifestyle habits and glucose management in one application, the *Diameter*. The core items to be measured by the *Diameter* are food intake, physical activity, glucose values, and medication use, with a built-in possibility to add other relevant items. This information is integrated in such a way that it gives individual patients and HCPs insight in lifestyle, blood glucose levels, as well as into the effect of lifestyle behaviour on glucose values in daily life, e.g., the HCPs can use the data about diet to gain insight in the (macro) nutrients intake and use this in their daily work to be able to treat the patients better. The lifestyle data can subsequently be used to give patients tailored coaching in order to improve adherence to lifestyle guidelines and, if necessary, medication use. The *Diameter* can be implemented as stand-alone application as well as in blended care forms, in which regular doctor visits are combined with online interventions. This paper describes the process of formulating the main requirements of the *Diameter* to measure dietary intake, physical activity, and glucose levels. The requirements for the evaluation of medication adherence and coaching will be addressed later.

## 2. Materials and Methods

We defined the requirements of the core items of the *Diameter* using an iterative mixed method design approach that combines data derived from several sources (Figure 1). Using the findings from the Diabetes and Lifestyle Cohort Twente (DIALECT), a literature search, and two pilot studies, the initial requirements were assessed. Subsequently, these preliminary requirements were discussed and are further elaborated upon in expert meetings to formulate the final set of requirements. The whole process is based on the approach of “the Function and events, Interactions and usability, Content and structure, and Style and aesthetics” (FICS) framework to enable the proper communication between researchers and the developers [23,24]. These requirements provide the foundation for the technology design of the *Diameter*, including what the system should do, which data is collected, what should be displayed, and what the user will experience. Using the iterative approach, the development runs parallel with the user experience, resulting in continuously made improvements that are based on feedback from both patients and professionals.

### 2.1. Cohort Study

As a first source of requirements, experiences in clinical practice were used to evaluate our methods for the data collection in patients that were included in the DIALECT-2 study. DIALECT-2 is a cohort study that was performed in ZGT, Almelo and Hengelo, the Netherlands, and it is designed to investigate the relationship between lifestyle parameters and long-term outcome in T2DM patients. As such, several parameters of interest for the *Diameter* are measured in DIALECT-2 [12]. The inclusion of DIALECT-2 (*n* = 400) is ongoing and is expected to be completed in 2020. In this study, for the diet, activity, and glucose part, 64, 98, and 60 patients were included from the DIALECT-2 cohort, respectively.

In DIALECT-2, a food diary was used to assess actual dietary intake. Towards this purpose, the patients were instructed to register the timing, amount, and type of all dietary intakes for two consecutive weeks. We have developed software that automatically calculates the intake of food components and macronutrients from these data, using an algorithm that was based on the Dutch Food Composition Table [25]. The entry of nutritional data is important for the *Diameter*, therefore the food diaries of 64 patients included in DIALECT-2 were analysed to evaluate the adherence of registration of the food intake. We evaluated adherence of registration for the respective meals during the day, and whether adherence changes during the two-week registration period. Furthermore, we investigated the daily distribution and between meal variability of registered carbohydrate intake, since this parameter is of particular importance in diabetes. The evaluation of adherence in registering food intake was based on three items: appropriate registration of meal records, appropriate registration of time records, and quantitative and qualitative description of content (e.g., 1 glass of milk), expressed as percentage of total meal moments, with 100% representing full compliance. Carbohydrate intake was calculated for the three main meal moments (i.e., breakfast, lunch, and dinner), the in-between meals, and the total intake in grams per day. Variability in carbohydrate intake for each meal was calculated using the within-person coefficient of variation (COV). Next, we calculated the number of days needed to get a representative estimation of a person’s true intake with a specified degree of error using the calculated COV [26].

Second, we assessed the requirements for the collection of data on physical activity. To this end, we evaluated the process of data collection on physical activity in the first 98 patients that were included in DIALECT-2. These patients wore a step counting device for one week (Fitbit^®^ Flex, Fitbit Inc., San Francisco, CA, USA) to measure physical activity and sedentary behaviour. The measurements with the activity trackers were analysed in terms of the use of activity sensors by patients, and activity behaviour in terms of activity bouts, sedentary behaviour, and sedentary bouts. The Fitbit has previously been validated for measuring step count during aerobe activities and for measuring sedentary behaviour [27,28,29]. Raw Fitbit data (steps/min) were organized into ready variables by an algorithm written in MATLAB^®^ (2016b, The MathWorks, Inc., Natick, MA, USA). We evaluated whether the patients experience limitations in the use of these devices and which are the technical limitations.

We used a flash glucose monitoring (FGM) device, the Freestyle Libre^®^ (Abbott Diabetes Care, Alameda, CA, USA), to evaluate the experienced limitations by the first 60 patients that were included in DIALECT-2 [30]. The Freestyle Libre sensor measures the average subcutaneous interstitial glucose level every 15 min throughout a two-week period. These glucose values, which serve as a surrogate for blood glucose, are transferred to a reader by the patients every 8 h. After the registration period of two weeks, the glucose data was derived from the reader, analysed by a MATLAB script, and then used for further analysis when ≥ 5 days of measurements, each with ≥ 90% available measurements, were available. Again, the experiences with the use of these kind of glucose sensors are translated into requirements for the *Diameter*.

### 2.2. Literature Search

We reviewed the existing mobile health (mHealth) applications that monitor dietary intake, physical activity, or glucose values to investigate the experiences with those applications and to determine the strength and limitations. There are many basic features in existing applications with undisputable value. We implicitly intend to use current standard features for the *Diameter* and will not formulate these as separate requirements. Rather, for the purpose of this study, we focus on functionalities that are either new or not yet standard.

A PubMed search was performed in the beginning of 2017 using a combination of the following search terms: “application”, “diabetes”, “diet”, “eHealth”, “food”, “food diary”, “mHealth”, “nutrition”, “nutrition diary”, “smartphone”, “physical activity”, “exercise”, “blood glucose”, or “continuous glucose monitoring”. A selection of relevant studies, which were performed in T2DM, was used to derive requirements for the *Diameter* based on the results.

### 2.3. Pilot Studies

We performed two pilot studies to gain insight into what people experience or want to experience using a mobile application. The aim of pilot study 1 was to identify the strengths and limitations of the state-of-the-art existing applications for diet registration. Therefore, in the beginning of 2017, the major mobile platforms (Apple iOS App Store, Google Play Store) were searched to find suitable diet registration mHealth applications. The following search terms were used (translated in Dutch): eating, calorie counter, food, food diary, nutrition, nutrition diary, and diabetes. mHealth applications with the following criteria were included: functionality on both Apple iOS and Android, at least fifty user reviews or ratings, capabilities for nutritional monitoring, the presence of a searchable nutritional database, the use of Dutch measurement units (e.g., gram, slices, cups), and the latest version of the application must have been released after 2014. Paid mHealth applications or those that were not available in Dutch were excluded [31]. For applications that met the criteria, all relevant information, such as the developer and the average user rating, was documented. This resulted in eight applications that were downloaded onto a smartphone by the researcher and then evaluated on certain key characteristics, including e.g., the use of the Dutch Food Composition Table as nutritional database [25], the possibility of manual entry of products, foods specificity (e.g., whole grain bread or white bread), and specific time of ingestion. Based on these criteria, the three most suited mHealth applications were selected for further testing by healthy volunteers for three days, in random order. After the test period, the usability of the mHealth applications was determined using a ten item-questionnaire that was based on the System Usability Scale (SUS) and the Unified Theory of Acceptance and Use of Technology (UTAUT) that was specifically designed for this purpose, and by two open questions to appoint the strong and weak characteristics [32,33,34]. The SUS assesses the usability and the UTAUT predicts the intention to use a technology [35].

In pilot study 2, the aim was to determine disease awareness in T2DM patients. To this end, we evaluated the awareness and knowledge of T2DM patients on the subject of healthy lifestyle and their illness. Additionally, we assessed patients’ requirements in future (coaching) technology that supports better diabetes management and healthy lifestyle choices [36,37].

### 2.4. Expert Meetings

During monthly expert meetings, the progress regarding the development of the *Diameter* was evaluated. The average number of attendants at these meetings was 10, including a wide array of experts, i.e., clinicians, professors in telemedicine, engineers, software developers, and researchers with expertise in (technical) medicine, nutrition, biomedical engineering, and computer science. During these meetings, the preliminary requirements that were derived from the cohort study, literature, and pilot studies were presented and discussed. Additionally, requirements were formulated during these meetings based on the expertise of the participants in the meetings. Taking existing applications regarding diet, physical activity, and glucose values as a starting point, the requirements that were distinctive as compared to the existing applications described in literature were formulated according to the FICS categories by a technical physician and then discussed with the experts. The requirements were prioritized with “must have”, “should have”, “could have”, or “wish to have” during brainstorms with the experts. This approach was used to ensure clear future communication with software developers regarding the required system functionality.

## 3. Results

The requirements that were derived from the cohort study, literature, pilot studies, and expert meetings are described below for the separate components (diet, physical activity/sedentary behaviour, glucose values) and as shared (i.e., applicable on all categories) requirements. The requirements are noted with identification (ID) in parentheses, coded using an F, I, C, or S to classify the requirements according to the FICS structure. An overview of the requirements, with corresponding ID numbers, can be found in the four Appendix A.

### 3.1. Requirement for Measuring Dietary Intake

#### 3.1.1. Cohort Study

We evaluated two-week food diaries in 64 patients who participated in DIALECT-2. The total carbohydrate intake of breakfast, lunch, dinner, and in-between meals contributed to 20.0%, 22.8%, 28.2%, and 29.0% of total intake, respectively. The COV was for each meal moment 45.0, 49.9, 53.4, and 70.2, respectively. Consequently, as the COV for carbohydrate intake for in-between meals was the highest, the in-between meals require the highest number of days (11 days) for a valid estimate within 30% of the true intake, higher than the number of days that are needed for breakfast (four days), lunch (eight days), and dinner (nine days) [38]. This information is necessary, because the compliance of registering diet decreased with in total 4.2% in the two-week period [26]. This indicates that, without further intervention, these patients will not register their diet properly during longer periods. Patients need to know what the minimum number of days is to enter their diet to be able to help them properly with their diabetes management (I1). This will help in motivating them to adhere. In addition, the *Diameter* should be intelligent and learn from the previously entered food items and use this to show smart options to ease dietary registration (F1). For example, it should inquire whether the patient ate their usual breakfast and, if not, provide the option to enter an alternative breakfast. For dinner, the system must automatically save different frequently used meals, and it should ask the patient whether they had eaten one of the saved meals, and, if not, have the option to change the saved meals (e.g., portion size). Secondly, to prevent the decline and insufficient registration of diet, the system must keep track of a personalized history per type of meal to prevent the patient from having to go over long lists of food options (F2).

In 41% of the patients, the overall description of food intake was not sufficient to draw meaningful conclusions, with the lowest compliance for dinner registration: Of the 64 patients, 26 were excluded due to too low compliance, and in 19 of these, this was due to inadequate description of the dinner [26]. In order to improve the overall adherence for diet registration, dinner entry in particular must be specifically designed to prevent underreporting. Therefore, we conclude that the system should split the dinner into main components, such as rice/potatoes/pasta, vegetables, meat/fish/meat substitute, gravy/sauce, and other (dessert/drinks etc.) (F3). Besides the data entry for dinner, the complete reporting of beverages is an item of concern. To reduce underreporting, we decided that the system must ask whether the patient drank something when food is entered without beverage (I2).

#### 3.1.2. Literature Search

The literature search resulted in a selection of seven articles, which were suitable to derive some important requirements for the *Diameter*. These articles report that the use of mHealth applications for monitoring nutrition results in barriers for both patients and clinicians [39,40]. Existing mHealth applications often only record the number of calories, whereas for diabetic patients, specific nutritional components have particular interest for blood glucose control, i.e., carbohydrates [41]. Specific information is necessary to demonstrate the impact of a specific type of food on the glucose levels [31,42]. This results in the requirements that the amount of carbohydrates must be displayed (F4) and the effect of carbohydrates or a specific type of food on the glucose value should be shown (F5). Patients also appreciate to see, in addition to the amount of consumed calories, the amount of calories that are left to eat that day and the amount of calories burnt by their physical activity (F6) [43]. Finally, healthy recipe suggestions should also be given (C1) [44].

#### 3.1.3. Pilot Studies

In pilot study 1, 20 healthy volunteers tested the usability of three existing mHealth applications; FatSecret, MyFitnessPal, and Virtuagym, for the monitoring of dietary intake. We demonstrated that improvements are necessary to implement a food tool in clinical practice. The usability study led to a number of requirements for the *Diameter*. Firstly, the time and date of ingestion should be registered (F7) before a meal or food product can be entered (S1), and the current date should be shown by default (S2). Furthermore, household measurements must be connected to the food product (F8), it should show pictures of food products (S3), it should be possible change or remove entered data (F9), it should be possible to add food products to a meal category, such as breakfast, lunch, and dinner (F10), a suggestion must appear when typing the first letters of a food product (I3), it must be made clear and easy to find the right food products (C3 and C4), an overview must be given of the consumed calories, amount of fat, carbohydrates, proteins, and percentage dietary reference intakes (F4 and F11), and often used food products must be remembered as is also described as result of the cohort study (F3). As addition to manual entry, having the optional use of a barcode scanner is often desired (F12), it should be possible to use the system on a website on the laptop or personal computer (I4), the application should send reminders or push messages to stimulate complete data input (I5), and it should provide short educational facts, e.g., explain what a calorie is (C5 and S4) [35].

#### 3.1.4. Expert Meetings

A main point, not yet addressed in the pilot, cohort, and literature study, was how to motivate patients to persistently use the application. Therefore, the focus of one expert meeting was on this subject. The attendants were challenged to make a quick design of a food registration tool with particular focus on attraction and ease of use, and this resulted in some requirements that aimed to make the tool interactive and visual. One requirement that followed was to display a graphical image of a plate, bowl, and a glass and to give the option to virtually drag food and drink to this plate and glass (S5). The idea behind this is that the amounts of food can be more easily determined when presented on the virtual plate. Also, there are applications in development that use photography technology to assess diet. This is a promising approach for entry food data, reducing the number of actions that are required to simply taking a picture of the meal [16,45]. The incorporation of nutrient estimation using photos taken with the mobile is desired (F13). People usually have a limited repertoire of meals, hence there must be an option to enter standard meals and to let the system remember earlier registered meals (F14), and there should also be the option to enter and save own recipes for re-use (I6). Fourthly, the system should ask smart questions (F15). For example, after initial inquiries on the regular use of milk and sugar in coffee, the system can ask, ‘did you add sugar as you usually do?’ when the user registers intake of coffee. This reduces the number of required actions, gives the impression of a personalized approach, and makes it interactive.

Other requirements that followed from the expert meetings are to store the data output per day and present it separately for breakfast, lunch, dinner, in-between meals, and total carbohydrate intake (F16). This gives a clear overview of the carbohydrate distribution over the day. Secondly, the application should be able to give a healthier option of a product that the patient entered (C6), e.g., whole grain bread instead of white bread. This gives the patient insight in healthier decisions. Thirdly, there must be an option to add new products to the database to keep the food database up to date (F17). Food products that are added by patients should be entered in a separate database. The products in this database can be checked on nutritional value and then added to the general database to give all users the ability to register this product. Fourthly, the app must contain a guideline with instructions to monitor food (C7). Fifthly, the data output is presented in gram per day. For each product, the amount of e.g., carbohydrates in gram per day should be calculated, based on nutritional values per 100 g of the food item, as noted in the Dutch Food Composition Table (C8). Finally, it should be possible to use the system independent of internet connection. At any time of the day, patients should have access to the food record to limit memory bias (I7).

### 3.2. Requirements for Measuring Physical Activity and Sedentary Behaviour

#### 3.2.1. Cohort Study

A Fitbit activity tracker was worn by 98 patients of the DIALECT-2 cohort until then to measure the number of steps per minute during one week. During these measurements we gained general insight in the applicability of such sensors in daily practice, but also the data on physical activity and sedentary behaviour of T2DM patients were helpful.

Regarding the applicability, the battery of the accelerometer needs to be charged by the user approximately every four to five days for 2 h. Some patients forgot to re-attach the sensor after charging. Some activities, like cycling, a common activity in the Netherlands, are not registered while using an accelerometer around the wrist. Based on these findings, the following requirements for the Diameter were formulated: The system must give a notification when the battery is empty (F18) and needs detection when the tracker is not worn to give a reminder to wear it (F19). Activities that a regular accelerometer around the wrist does not detect must also be measured by using an application on the mobile device that detects cycling (F20) or by the manual insertion of non-recordable activities (I8).

Regarding physical activity, only a few patients met the criteria of an activity bout, which is an activity of at least 10 consecutive minutes of moderate to vigorous physical activity (MVPA) with ≥ 95 steps/min [46,47]. About two-thirds of the patients (69%) had no bout of MVPA at all in seven days, whereas ≥ 150 min MVPA per week is recommended, e.g., to reduce the risk of cardiovascular diseases [5,6,48]. Nevertheless, the vast majority (93%) were able to achieve the intensity of moderate activity at one point during the follow-up time, however the duration of the activity was too short [49]. The *Diameter* should provide education for patients regarding activity bouts and physical activity (C9), in order to stimulate patients to achieve MVPA bouts, which are beneficial for their diabetes regulation [42,44]. Also, the application should detect when the intensity of MVPA or duration of an activity bout is not met, in order to provide the basis for the future coaching module to be developed (F21).

Independent of the amount of MVPA, sedentary behaviour increases the risk of morbidity and mortality [50,51,52,53], and it is recommended that sedentary bouts are be interrupted every 30 min with an activity of light intensity of more than 10 steps/min for at least 1 min [48,54]. Our data demonstrated that our T2DM patients had no movement for 76% of the total waking hours per day, of which 7 h were spent in prolonged sedentary bouts of at least 30 consecutive minutes [49]. We therefore want the system to create awareness regarding sedentary behaviour and motivate patients to minimize sedentary time (C10).

#### 3.2.2. Literature Search

There are various studies addressing the issue of evaluating physical activity while using mHealth applications. An important finding was that patients prefer the visual demonstration of their activities, by, for example, bar charts over merely numerical presentation (S6) [55]. Also, patients liked to be reminded when they were inactive for a prolonged time period (I9) [43]. Literature also confirmed the notion of paying attention to the management of body weight management, being closely related to physical activity and diet. Weight loss strongly contributes to improved glycaemic control. It is therefore necessary to show patients the burned calories during a performed activity (C11) and to have an option to register body weight (I11) to follow up on weight (loss) in time (C12) [11,44,48,50]. One of the aims in lifestyle management is to reduce sedentary behaviour. The *Diameter* should therefore detect sedentary periods (I11) and it must give educational information regarding sedentary behaviour (C10), as also described in the cohort study [39,56].

#### 3.2.3. Expert Meetings

During the expert meetings, some additional requirements concerning physical activity and sedentary behaviour were formulated: First, there is a strong preference for the option to connect multiple types of activity sensors (I12). Secondly, the system must start with measuring of a baseline to determine the current activity and sedentary behaviour of the patient (F22). When, in the future, a coaching module is added in the *Diameter*, these baseline data should be available.

### 3.3. Requirements for Measuring Glucose Values

#### 3.3.1. Cohort Study

In the DIALECT-2 cohort, we performed flash continuous glucose measurements in 60 patients during two weeks. Following, we gained insights in the applicability of FGM sensors, the level of glycaemic control, and the daily glucose variability between patients, which are used to formulate requirements.

First, we investigated the usage of the sensor. Of the 60 patients, 12 patients had to be excluded because there were less than five days in which at least 90% of data were available. The most important causes for data loss were the patient forgetting to scan the sensor by the patient and the premature loss of the Freestyle Libre sensor. Of the latter, in 35% of the cases, detachment of the sensor was the problem, 29% of which occurred in the first week. In seven patients, a new sensor was attached. The system must give notifications to reduce data loss by not scanning the sensor (I13) and it should have the option to connect a new Freestyle Libre sensor (I14).

Regarding the glucose level of these patients, the level is, on average, 58.6% of the time that patients are in between the glucose target ranges (≥ 4 mmol/L and ≤ 8.4 mmol/L). The remainder of the time they have hypoglycaemia (4.2%) or hyperglycaemia (37.2%). Patients need the insight in their glucose values, including the percentage of hypo-, hyper-, and normoglycaemic episodes, throughout the day and the progression over time to be better capable in keeping the values between the target ranges (F23). Besides that, they need insight in their glucose variability (F24). The patients with high glucose variability need to have these insights to be able to act better on the glucose values and decrease the variability. To be able to do this, education is needed regarding glucose targets, glucose variability, and on the unfavourable long-term effects of uncontrolled hyperglycaemia (C13).

#### 3.3.2. Literature Search

There are several glucose sensors on the market that are used for continuous glucose monitoring (CGM) in the clinical practice of diabetes management. These so-called real-time CGM sensors are usually used in conjunction with a subcutaneous insulin pump and the use is labour intensive for the patient, because twice daily calibration by finger pricks is still needed, rendering such sensors unsuitable for the purpose of the *Diameter* [57]. We therefore decided to choose the only alternative currently available, i.e., the Freestyle Libre system. It is obvious that using CGM or FGM provides detailed information regarding glycaemic control, with the additional advantage of wireless transfer of the glucose data (I15) [55,58]. For the *Diameter*, these data are to be used to display glucose trends (F23) and to provide the patient and HCP with insight about blood glucose levels and glucose variability (C13), as also described in response to the cohort study. This helps to effectively engage patients and give them the insight that they need [44,58]. The data should be displayed in an easy understandable graph that shows how physical activity and nutrients affect blood glucose (S7) [42].

#### 3.3.3. Expert Meetings

Additional requirements concerning blood glucose were formulated based on the expert meetings. First, the target range for blood glucose should be adjustable to allow for personalized targets (F25). In addition, there is the wish for an option to warn when the glucose values are out of range. (I16). Secondly, it is desired that the *Diameter* can synchronise with insulin pumps and also with other glucose measuring devices than the FGM currently used (F26). This includes the option of manually inserting measured glucose values by a finger-prick (I17). Thirdly, the systems should allow for the development of algorithms that predict glucose levels based on the measured blood glucose data, physical activity data, and food intake data (C14).

### 3.4. Shared Requirements

#### 3.4.1. Literature Search

Beside requirements for the separate items to be measured, we also derived some general shared requirements from the literature for the *Diameter*. The lack of coordination with the HCPs is a main issue with currently available lifestyle applications. Data that are collected with such applications by an individual are not available for the clinicians, and certainly not in an organized fashion. We want to develop an application that allows for connecting and sharing the data with the care provider (I18) (I19). This can be achieved by the option to automatically generate a report of the data, which is digitally sent to the HCP. The communication and information exchange should contribute to better decision-making [55,58]. Other important requirements are related to the convenience of use. To this purpose, the language and tone should be accessible, encouraging, and supportive (C15) [59], generally intelligible symbols and terms should be used (S8) [59], and the necessity for scrolling must be minimized (S9) [44]. In addition, medical terminology must be used where needed, but clear explanations should be provided (C16) [44]. Also, keeping the necessary active use to less than 15 min a day is desired (I20) and the applications must be provided in the native language of the user (C17), resulting in the persistent use of the app [60]. Finally, costs appears to be a significant concern, with most people being unwilling to pay anything for apps and discontinuing use when they find that in-app payments are required, therefore the aim is to provide the app for free (I21) [43].

#### 3.4.2. Pilot Studies

In pilot study 2, 19 patients participated in the questionnaires and interviews about awareness and a technology to support diabetes management. Firstly, the patients were asked questions about the following subjects to examine their overall awareness: influence of nutrition and exercise on glycaemic control, diabetes complications, self-management of the respondent, and desirable behaviour on exercise, nutrition, and weight. Approximately 40% of the patients correctly answered half of the questions regarding overall awareness. However, most of the respondents did not have the supposed knowledge regarding the effect of exercise and nutrition on glycaemic regulation. For example, only a few patients were aware of the effect of even a small percentage of weight loss on the improvement of the blood glucose level. The level of awareness and the interviews held with the patient, resulted in requirements to give the patients more knowledge and insight, as noted earlier (F4, F6, F23, C5, C9, C10, C13, S4). New requirements that were mentioned in the interviews were to also incorporate blood pressure measurements (I22) and incorporate an insulin bolus calculator, i.e., a calculator to determine the appropriate insulin dose before the meal, based on the current glucose level and the amount of carbohydrates in the meal (F27) [36].

#### 3.4.3. Expert Meetings

In the expert meetings, it was brought up that patients may have a preference to focus on specific lifestyle items and this may change in time. For example, the aim may be to get more active, instead of focusing on healthy diet. It should therefore become possible to use the application selectively to prevent overload with excessive information (I23). Also, data must be stored according to the European privacy laws (F28). Thirdly, the data must be stored for at least one year (C18). Fourthly, the clocks of all different sensors must be synchronized (F29) to ensure that no discrepancies in time can occur. Fifthly, an overview of the data in the past can be found in an overview per day, per week, and per month (S10). This gives the ability to look at trends. Finally, all of the patients are able to enter all desired data without assistance, independent of education level (I24).

### 3.5. Overview of the Requirements

The requirements are organised according to the FICS framework and labelled as “must have”, “should have”, “could have”, and “wish to have”. In total, 81 requirements were formulated, which are of added value when compared to current applications. Of these, 29 are formulated as functional requirements, 24 as interactive and usability requirements, 18 as substantive and structural, and 10 as style, as can be seen in Table 1. Of these requirements, 74% were labelled as “must” and “should” have and 26% as “could” and “wish to” have. The requirements were labelled during the expert meetings. The “could” and “wish to” have requirements were labelled in this category, because they were valued as requirements that are not of utmost importance in this stage of the development of the app, to use the app, or to receive the desired information and insight. However, these requirements have the potential to increase the ease of use or have the potential to motivate to use the app more.

## 4. Discussion

We are developing a digital tool, the *Diameter*, with the aim of improving adherence to lifestyle in patients with T2DM. The tool collects and integrates information of diet, physical activity, and glucose values, i.e., items that are pivotal for the management of patients with T2DM. In order to formulate the requirements for this tool, we applied a mixed method design approach in which we used experiences from large scale data collection in a cohort study, performed a literature search, performed pilot studies, and organized expert meetings.

Although many applications are already available to assist patients in monitoring lifestyle behaviour like their diet and physical activity, these applications are not designed for the follow-up of chronic diseases, like diabetes. Therefore, they lack integration with glucose measurements and also do not allow blended care, i.e., function in connection with HCPs [39,40,55,61,62]. Due to the tight relationship between lifestyle and glucose management, an integrated approach using these key elements, together with a blended care setting, were starting points for us. We evaluated existing applications for lifestyle and T2DM to determine the requirements for each part of the *Diameter*. Of these applications, 15 requirements were of value and were therefore adopted. Additionally, 49 new requirements were formulated for the diet, physical activity, and glucose value part during the cohort study, pilot studies, and expert meetings. Besides, shared requirements that are of importance for the integration of the components also had to be formulated, resulting in another 17 shared requirements.

To achieve adherence to lifestyle advice in diabetes, it is important that patients have knowledge on how lifestyle behaviours affect their condition. The pilot study concerning awareness showed there is an urge to improve knowledge of patients on the effects of carbohydrates and activity on the glucose values. We expect that demonstrating the effects that food choices directly have on glucose level will provide a very strong feedback mechanism that may help to stimulate healthy behaviour. The necessity for lifestyle interventions is emphasized by the results of our cohort study that there is a lot of room for improvement on diet, physical activity, sedentary behaviour, and glucose values. Therefore, 14 requirements, mostly labelled as content and structure and style and aesthetics, of the *Diameter* are related to providing insight in counting carbohydrates, glucose values, physical activity, and how these factors mutually influence each other.

Almost half of the requirements, 37 out of 81, are formulated for diet registration. From our findings in the cohort study and the pilot study it was clear that, in order to gain complete and reliable entry of dietary information, the functionality of a new nutrition entry tool needs substantial improvement when compared with existing diet applications. The existing applications require frequent manual data entry and the process of data entry is relatively time consuming. The requirements defined to solve these issues relate to incorporating smart options, mostly labelled as function and events requirements. E.g., the application must learn from past entered data and the use this information to ask personalized questions and to give personalized suggestions in the future. Another functional requirement to increase the ease of use was to enter a main meal by using pre-defined components.

The integration of smart options is important, not only for the dietary entry, but also in the other components of the *Diameter*. For example, the application must recognize when the glucose sensor needs to be scanned, when the patient has not worn the activity tracker for too long, and it must detect activities that the activity tracker cannot measure.

Incorporating the right interaction and usability requirements are also of importance in making the application more user-friendly and to stimulate maintenance of use. Examples of such requirements are the possibility to enter own recipes, the possibility to use the app without access to the internet, to allow own choice of activity tracker and glucose sensor, to enable potential connection with a (digital) healthcare professional, and individual choice for which educational modules are switched on/-off.

As stated above, providing insight in the effect of diet and activity on the glucose values is a key element of the *Diameter*. To this end, the application can generate (e.g., past day, week or month) a graphical presentation of these data and can recognize and show trends for different time periods. Reports of the data can be shared with the diabetes professional, allowing for blended care.

The strengths of this study are that the requirements were developed from an integration of four approaches (cohort study, literature search, pilot studies, and expert meetings), that the requirements are developed in an iterative process, and the main components for T2DM treatment are taken into account. However, there are a few sources of potential bias. There could be selection bias, because the requirements were developed from a complicated T2DM population, which was located in a specific region in the Netherlands. It is possible that this resulted in other requirements than we would have found by researching a population less complicated or with different ethnical background. Also, it is possible that a different composition of the group of experts would have led to other requirements due to differences in personal experience and specific knowledge. However, when expanding to other populations, the iterative approach of the development process should circumvent these potential sources of bias.

With the formulation of the initial requirements that are described in this paper, the first step of the development of the *Diameter* has been taken. From here, we intend to let patients use a first version of the Diameter in a new pilot study. The data generated in this pilot study will serve several purposes. First, an enhanced set of requirements will be derived. We also intend to formulate requirements for the monitoring of adherence to pharmacological therapy. Also, in a parallel process, the data will be used as input for the development of the coaching part of the *Diameter* to be developed. The coaching part will be designed to provide data-driven tailored coaching, based on inter alia, individual patient data, preferences, and comorbidities in order to achieve and maintain adherence to lifestyle recommendations.

In summary, the development of the *Diameter* is an iterative process, using a multi-method approach. Every time after introducing a new version, a pilot study will be performed to evaluate the app in terms of effectiveness, acceptance, and feasibility, and to fine-tune the requirements.

## 5. Conclusions

The development of a new tool for patients with T2DM is important, because insight in their diet, activity, and glucose values is currently lacking. This study describes the development of the requirements for the first version of the *Diameter*, which are focused on gathering the necessary data and giving patients insight. For the development of future versions of the Diameter, in which a tailored data-driven coaching module will be incorporated inter alia, it is an important step to be able to efficiently collect lifestyle and glucose data.

Future research is needed to develop the desired application for the patients to receive tailored coaching, blended in their healthcare. This future research will focus on development of the tailored coaching module and algorithms that take into account the duration of diabetes, comorbidities, and personal preferences. Also, research is necessary to optimize the usability of the application for patients and HCPs by evaluating experiences of different diabetes populations (e.g., first line and second line healthcare) with the app, evaluating feasibility of the application integrated in clinical care, and by optimizing adoption and implementation of the application.

## Figures and Tables

**Figure 1 nutrients-11-00409-f001:**
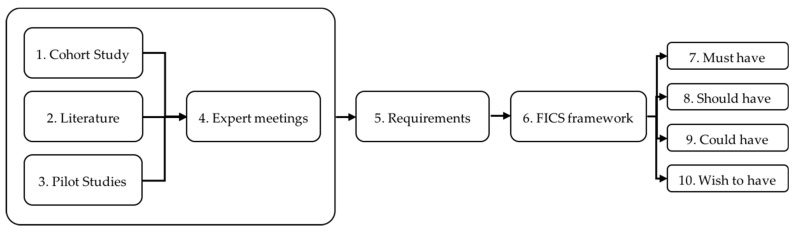
Requirements were formulated from insights gathered in the Diabetes and Lifestyle Cohort Twente (DIALECT) cohort study (**1**), literature research (**2**) and pilot studies (**3**). In expert meetings (**4**) these requirements were discussed and new requirements (**5**) were added. The requirements were formulated according to the Function and events, Interactions and usability, Content and structure and Style and aesthetics (FICS) framework (**6**). The requirements were labelled during expert meetings as “must” (**7**), “should” (**8**), “could” (**9**), and “wish to” (**10**) have.

**Table 1 nutrients-11-00409-t001:** The number of formulated requirements as a result of the cohort study, literature search, pilot study, and expert meetings per FICS category for diet, physical activity (PA), and sedentary behaviour (SB), glucose values and shared.

	Diet (*n*)	PA and SB (*n*)	Glucose Values (*n*)	Shared (*n*)	Total (*n*)
Function & events	17	5	4	3	29
Interaction & usability	7	5	5	7	24
Content & structure	8	4	2	4	18
Style & aesthetics	5	1	1	3	10
	37	15	12	17	81

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
