# Peer review of "Requirements of an Application to Monitor Diet, Physical Activity and Glucose Values in Patients with Type 2 Diabetes: The Diameter"

_nutrients, 2019, doi:10.3390/nu11020409_

Round 1
Reviewer 1 Report
The paper deals with an interesting development of tools for managing health-related aspects and therapies particularly type 2 diabetes.
The study presents several aspects of interest however, a better description of the rationale for the aim and for the chosen methodology would be helpful.
What is not clear is why developing a new mHealth device when several others have been developed according to the Authors.
Detailed comments are reported below
- Introduction
References to the European eHealth policy and the mHealtj promotion by the World Health Organization (WHO) are missing.
https://ec.europa.eu/digital-single-market/en/european-policy-ehealth
https://www.itu.int/en/ITU-D/ICT-Applications/eHEALTH/Be_healthy/Pages/The-EU-mHealth-Hub-Project.aspx
https://www.ehealth-hub.eu/
https://www.who.int/ncds/prevention/be-healthy-be-mobile/REOI_hub_ppt_17August2017.pdf?ua=1
A list of European projects concerning eHealth and mHealth.
PREventive Care Infrastructure based On Ubiquitous Sensing (PRECIOUS) http://www.thepreciousproject.eu/
Quality Information Services and Dietary Advice for Personalized Nutrition in Europe http://www.qualify-fp7.eu/
PD_Manager: mHealth platform for Parkinson's disease
https://ec.europa.eu/programmes/horizon2020/en/news/pdmanager-mhealth-platform-parkinsons-disease
- Methods
The reader can guess in this way all variables and functions are considered but it should be deepened. Particularly, Authors say “ Whereas a food frequency questionnaire was used in DIALECT-1, in DIALECT-2 we extended the data collection with a food diary.” The two data collection technique have different aims, the one allows for assessing habitual intake over a period of time, the second one allows for assessing actual intakes and the estimates of habitual intake can be derived applying specific statistical tools. The food frequency questionnaire or food propensity questionnaire provide inputs to the procedure to estimate the usual diet. So, the food diary cannot be considered an extension of the food frequency questionnaire. Maybe the wording should be modified.
The Authors say “A selection of relevant studies, performed in T2DM, was used as it was not the intention to perform a systematic literature review, but merely to derive requirements for the Diameter based on the results.” The question is how did they derived the requirements if the systematic review was not performed? Please, explain.
In the pilot study 3 mHealth dietary assessment applications were selected from literature (rows 161-164), downloaded on devices and then distributed to professionals “and three mHealth applications were selected for further testing. Does this mean that there were more than three applications? This seems to contradict the sentence in rows 161-164. Please, clarify.
Row 172. Was the quality of food composition tables considered other than the wideness?
Moreover, the adopted food classification system was evaluated or considered?
Which methodology was adopted to finalise the experts meetings?
Finally, how had been the numbers of patients examined from DIALECt-2 (64 for dietary assessment 98 for physical activity) determined ?
- Results
Some parts are more related to the introduction rather than the description of the results. These are the cases of rows 197-202, 205, 235-238, 248-249.
The 3.1.4 Paragraph seems a discussion rather than a result.
3.5 Paragraph: the Authors say 74% of the items were labeled as “must” and “should have” and 26% as “could”. This seems an important results in characterising the tool, but the items are not evidenced. I would prefer a longer description here.
General question, why results from DIALECT-1 are not presented?
- Discussion
A comparison with other tools to stress the novelty of the Diameter would help in understanding why was the tool developed and the advantage of using it rather than other mHealth devices.
Do you consider the complementarity with other devices like scales (see e.g., https://www.teknoscienze.com/tks_article/innovative-pocket-size-bluetooth-kitchen-scale/)
- Conclusion
Conclusions are not provided on the tool but on the process to design the tool and the process to submit the new releases.
Author Response
Response to Reviewer 1
Comments to the Author
The paper deals with an interesting development of tools for managing health-related aspects and therapies particularly type 2 diabetes.
The study presents several aspects of interest however, a better description of the rationale for the aim and for the chosen methodology would be helpful.
What is not clear is why developing a new mHealth device when several others have been developed according to the Authors.
Minor points and discussion:
Introduction
Point 1: References to the European eHealth policy and the mHealth promotion by the World Health Organization (WHO) are missing.
https://ec.europa.eu/digital-single-market/en/european-policy-ehealth
https://www.itu.int/en/ITU-D/ICT-Applications/eHEALTH/Be_healthy/Pages/The-EU-mHealth-Hub-Project.aspx
https://www.ehealth-hub.eu/
https://www.who.int/ncds/prevention/be-healthy-be-mobile/REOI_hub_ppt_17August2017.pdf?ua=1
A list of European projects concerning eHealth and mHealth.
PREventive Care Infrastructure based On Ubiquitous Sensing (PRECIOUS) http://www.thepreciousproject.eu/
Quality Information Services and Dietary Advice for Personalized Nutrition in Europe http://www.qualify-fp7.eu/
PD_Manager: mHealth platform for Parkinson's disease
https://ec.europa.eu/programmes/horizon2020/en/news/pdmanager-mhealth-platform-parkinsons-disease
Response 1: Thank you for noticing that we indeed did not refer to these European mHealth and eHealth projects. We agree that taking advantage of the knowledge coming from these projects add to the paper. As such we added a new sentence in the introduction and included some of these references (row 60). In addition, some other studies of these projects are added as reference in other parts of our manuscript (row 62, row 72 and row 276).
Methods
Point 2: The reader can guess in this way all variables and functions are considered but it should be deepened. Particularly, Authors say “ Whereas a food frequency questionnaire was used in DIALECT-1, in DIALECT-2 we extended the data collection with a food diary.” The two data collection technique have different aims, the one allows for assessing habitual intake over a period of time, the second one allows for assessing actual intakes and the estimates of habitual intake can be derived applying specific statistical tools. The food frequency questionnaire or food propensity questionnaire provide inputs to the procedure to estimate the usual diet. So, the food diary cannot be considered an extension of the food frequency questionnaire. Maybe the wording should be modified.
Response 2: We thank you for this comment. Indeed, the food diary cannot be considered as an extension of the food frequency questionnaire. In DIALECT-1 we mainly used questionnaires, e.g. the food frequency questionnaire. In DIALECT-2 we also used these questionnaires, but extended the data collection with continuous measurements, e.g. physical activity and glucose values, and a food diary, indeed as suggested by this reviewer, to be able to assess actual intakes. In this manuscript we only formulated requirements based on the data collected in DIALECT-2. Therefore, we modified the wording to make clear that we only used DIALECT-2 to formulate the requirements. The food frequency questionnaire is not used in this manuscript to formulate requirements because we needed the actual intake to be able to formulate requirements and therefore has now been omitted to avoid misunderstandings.
Point 3: The Authors say “A selection of relevant studies, performed in T2DM, was used as it was not the intention to perform a systematic literature review, but merely to derive requirements for the Diameter based on the results.” The question is how did they derive the requirements if the systematic review was not performed? Please, explain.
Response 3: We understand that this sentence raises this question. A systematic review was not performed for several reasons: First There are only very few studies that have the primary aim to define requirements. Second: Most studies about the evaluation of an application are of interest but these lead to new requirements that are often mentioned in the discussion. For this reason, we choose to work with a selection of relevant studies.
Point 4: In the pilot study 3 mHealth dietary assessment applications were selected from literature (rows 161-164), downloaded on devices and then distributed to professionals “and three mHealth applications were selected for further testing. Does this mean that there were more than three applications? This seems to contradict the sentence in rows 161-164. Please, clarify.
Response 4: Thank you for pointing out this ambiguity. This was not noted correctly in the manuscript. The applications were selected from the major mobile platforms using a number of general inclusion criteria (row 167-171). This resulted in eight applications. After this first selection these applications were evaluated on a number of more specific characteristics (row 174-176). Three applications fitted best and were selected to test in healthy volunteers. We rewrote this section in the manuscript to make this clearer.
Point 5: Row 172. Was the quality of food composition tables considered other than the wideness?
Response 5: We selected the applications based on the use of the Dutch Food Composition Table, which is considered as a database with good quality and wideness. We rewrote this part in the manuscript to make clear these applications used the Dutch Food Composition Table as database.
Point 6: Moreover, the adopted food classification system was evaluated or considered?
Response 6: We did not evaluate the adopted food classification system, because of the use of the Dutch Food Composition Table as used database, which is an evaluated food classification system.
Point 7: Which methodology was adopted to finalise the experts meetings?
Response 7: We agree that it was not clearly written how the requirements were formulated according to the FICS categories and how they were prioritized. We added text to make this part clearer.
Point 8: Finally, how had been the numbers of patients examined from DIALECT-2 (64 for dietary assessment 98 for physical activity) determined?
Response 8: The development of the Diameter is an ongoing iterative process. To be able to develop the requirements of the first version of the Diameter, as described in this manuscript, different studies have been performed using the data gathered in DIALECT-2. These studies were performed with time limitations in which the data was used of included patients up to that moment, resulting in in different numbers of patients. As the purpose of the formulation of the requirements was not to find statistical significance, but to be able to develop a tool, it was also not of importance to calculate the number of patients necessary for the studies. As DIALECT-2 is still ongoing, more evaluations on the DIALECT-2 cohort are planned to follow.
Results
Point 9: Some parts are more related to the introduction rather than the description of the results. These are the cases of rows 197-202, 205, 235-238, 248-249.
Response 9: We understand that some of these parts read more as related to the introduction than description of the results. We adapted some of the parts which are indeed not necessary to point out in the results section as is the case for the first line of section 3.1.2. However, we kept all section headers and the first section of the results, as a guidance through the results section for the reader.
Point 10: The 3.1.4 Paragraph seems a discussion rather than a result.
Response 10: We agree with the reviewer that it seems like a discussion. However, this paragraph is part of the results. Therefore, we changed the wording to make the text fit in the results section.
Point 11: 3.5 Paragraph: the Authors say 74% of the items were labeled as “must” and “should have” and 26% as “could”. This seems an important result in characterising the tool, but the items are not evidenced. I would prefer a longer description here.
Response 11: Thank you for this comment. This part could be more explained. We added a longer description to make clear how the choice was made to label the requirements.
Point 12: General question, why results from DIALECT-1 are not presented?
Response 12: As mentioned in question 2 of the reviewer, in DIALECT-1 we did not collect continuous data. The continuous data collection of physical activity and glucose values and the actual food intake data collection was added in DIALECT-2. This information was necessary to be able to formulate the requirements for the Diameter. DIALECT-1 was necessary to get the insight about where the diabetes population needs support, shortly described in the introduction. We adapted this in the methods to avoid misunderstandings.
Discussion
Point 13: A comparison with other tools to stress the novelty of the Diameter would help in understanding why was the tool developed and the advantage of using it rather than other mHealth devices.
Response 13: A general comparison with other tools is made in the discussion. Compared to other applications, no application exists which integrates lifestyle with glucose values and with which it is possible to get coaching and use it blended in healthcare. The Diameter is still in development and this manuscript describes which requirements for the first version as such the focus in the discussion is more on the general comparison with existing apps instead of specific other mHealth devices.
Point 14: Do you consider the complementarity with other devices like scales (see e.g., https://www.teknoscienze.com/tks_article/innovative-pocket-size-bluetooth-kitchen-scale/)
Response 14: During the expert meetings we discussed complementarity of a various amount of devices. E.g. one of the requirements in the manuscript, I12, describes the requirement to connect multiple types of activity sensors. However, our idea is not limited to only the addition of activity sensors. If the patient wants to use e.g. a smart (kitchen) scale and connect this to the Diameter, this should be an option for this patient. However, as the focus of this manuscript is more on the physical activity, diet and glucose values, the summation of all possible devices was beyond of the scope of this manuscript.
Conclusion
Point 15: Conclusions are not provided on the tool but on the process to design the tool and the process to submit the new releases.
Response 15: The manuscript ends with a summary about the process to design the tool. A conclusion has been added which addresses the conclusion on the tool.

Reviewer 2 Report
General comments
1. This paper aims to develop a digital tool which incorporates lifestyle habits and glucose management, the Diameter. The core items measured by the Diameter were food intake, physical activity, glucose values and medication use and the authors argue that there is a built-in possibility to add other relevant items.
2. The requirements of the core items of the Diameter were defined using an iterative mixed method design approach that combines data derived from several sources: cohort study, pilot studies, literature search and expert meetings. The main cohort that was used to assess the initial requirements was from the Diabetes and Lifestyle Cohort Twente (DIALECT), a literature search and two pilot studies.
3. The authors rightly highlight the need for such an application in diabetes control as the use of technology can help to incorporate effective lifestyle management in routine clinical care. They also argued on the novelty of their technology in that existing applications are usually developed for personal use rather than for clinical use, no application exists that can measure and integrate all the information considered necessary for an optimal diabetes management, and coaching functionalities.
4.These results should be put in context owing to clarifications on the methodology used and the assumptions made in the analysis.
5. The authors need to clarify the robustness of the method used and assumptions made for the requirements of the core items of the Diameter. My specific comments are as follow:
a) Since the main cohort that was used to assess the initial requirements was from the Diabetes and Lifestyle Cohort Twente (DIALECT), how is the selection of this population is likely to influence the general population of diabetes? For generalization of the results, it is important to know the authors’ assumptions on the selected population and how this may affect other diabetes population?
b) in other words, different population may result in variations given differences of the population.
b) T2DM patients often co-habit with multiple diseases (co-morbidities). Did authors consider co-morbidities and how this will affect Diameter?
c) The requirements of the core items of the Diameter were formulated without patients or patient group inputs. Authors considered expert views during the expert meetings. Please clarify why patients or patient groups views were not considered and what is the likely impact of this?
Author Response
Response to Reviewer: 2
Comments to the Author
This paper aims to develop a digital tool which incorporates lifestyle habits and glucose management, the Diameter. The core items measured by the Diameter were food intake, physical activity, glucose values and medication use and the authors argue that there is a built-in possibility to add other relevant items.
The requirements of the core items of the Diameter were defined using an iterative mixed method design approach that combines data derived from several sources: cohort study, pilot studies, literature search and expert meetings. The main cohort that was used to assess the initial requirements was from the Diabetes and Lifestyle Cohort Twente (DIALECT), a literature search and two pilot studies.
The authors rightly highlight the need for such an application in diabetes control as the use of technology can help to incorporate effective lifestyle management in routine clinical care. They also argued on the novelty of their technology in that existing applications are usually developed for personal use rather than for clinical use, no application exists that can measure and integrate all the information considered necessary for an optimal diabetes management, and coaching functionalities.
These results should be put in context owing to clarifications on the methodology used and the assumptions made in the analysis.
The authors need to clarify the robustness of the method used and assumptions made for the requirements of the core items of the Diameter. My specific comments are as follow:
Point 1: Since the main cohort that was used to assess the initial requirements was from the Diabetes and Lifestyle Cohort Twente (DIALECT), how is the selection of this population is likely to influence the general population of diabetes? For generalization of the results, it is important to know the authors’ assumptions on the selected population and how this may affect other diabetes population? In other words, different population may result in variations given differences of the population.
Response 1: Thank you for notifying this potential issue. To be able to derive our requirements for the first version of the Diameter we used the results from studies performed in a complicated diabetes population treated in a hospital setting. It is possible that this resulted in other requirements than we would have found using studies from a less complicated diabetes population. However, finally it is the intention that the Diameter can be used by the whole diabetes population and as such will be built as flexible as possible and as a personalized tool. As the development is an iterative process, we will evaluate the Diameter also in other diabetes population which may led to new requirements we currently did not come up with.
Point 2: T2DM patients often co-habit with multiple diseases (co-morbidities). Did authors consider co-morbidities and how this will affect Diameter?
Response 2: This is a justified comment of the reviewer. An important part of the Diameter is the personalization of the application. Although not feasible in the first version of the Diameter it will be integrated in next versions. We added this in the discussion to address the vision of the Diameter better. This does not mean that patients with comorbidities cannot use the first version of the Diameter. These patients also benefit from insights in their food, activity and glucose values.
Point 3: The requirements of the core items of the Diameter were formulated without patients or patient group inputs. Authors considered expert views during the expert meetings. Please clarify why patients or patient groups views were not considered and what is the likely impact of this?
Response 3: These requirements are formulated using literature, cohort studies, pilot studies and expert meetings to know which requirements need to be incorporated to be able to develop the first version of the Diameter. Some patients were already involved in the development of the requirements as described in pilot study 2 in this manuscript. With the total number of requirements that we formulated, we can develop the design of the application and a working application itself. The next step will be to evaluate the Diameter in patients. This can result in new requirements which will be adopted in a new version of the Diameter. Again, the renewed version can be evaluated by patients. In short, the development of the Diameter is an iterative process in which we will incorporate feedback of the patients in future versions.

Round 2
Reviewer 1 Report
Teh paper has been enhaced claryfing the concepts, adding references, including a conclusion.
Author Response
Dear reviewer,
Thank you for your positive appreciation and approval of our response to your first constructive feedback.
Yours sincerely,
Niala den Braber